# Machine Learning Prediction Model of Tuberculosis Incidence Based on Meteorological Factors and Air Pollutants

**DOI:** 10.3390/ijerph20053910

**Published:** 2023-02-22

**Authors:** Na Tang, Maoxiang Yuan, Zhijun Chen, Jian Ma, Rui Sun, Yide Yang, Quanyuan He, Xiaowei Guo, Shixiong Hu, Junhua Zhou

**Affiliations:** 1The Key Laboratory of Model Animals and Stem Cell Biology in Hunan Province, School of Medicine, Hunan Normal University, Changsha 410013, China; 2Changde Center for Disease Control and Prevention, Changde 415000, China; 3Hunan Provincial Center for Disease Control and Prevention, Changsha 410005, China

**Keywords:** tuberculosis, machine learning, forecasting, neural networks, random forest regression, support vector regression

## Abstract

Background: Tuberculosis (TB) is a public health problem worldwide, and the influence of meteorological and air pollutants on the incidence of tuberculosis have been attracting interest from researchers. It is of great importance to use machine learning to build a prediction model of tuberculosis incidence influenced by meteorological and air pollutants for timely and applicable measures of both prevention and control. Methods: The data of daily TB notifications, meteorological factors and air pollutants in Changde City, Hunan Province ranging from 2010 to 2021 were collected. Spearman rank correlation analysis was conducted to analyze the correlation between the daily TB notifications and the meteorological factors or air pollutants. Based on the correlation analysis results, machine learning methods, including support vector regression, random forest regression and a BP neural network model, were utilized to construct the incidence prediction model of tuberculosis. RMSE, MAE and MAPE were performed to evaluate the constructed model for selecting the best prediction model. Results: (1) From the year 2010 to 2021, the overall incidence of tuberculosis in Changde City showed a downward trend. (2) The daily TB notifications was positively correlated with average temperature (r = 0.231), maximum temperature (r = 0.194), minimum temperature (r = 0.165), sunshine duration (r = 0.329), PM_2.5_ (r = 0.097), PM_10_ (r = 0.215) and O_3_ (r = 0.084) (*p* < 0.05). However, there was a significant negative correlation between the daily TB notifications and mean air pressure (r = −0.119), precipitation (r = −0.063), relative humidity (r = −0.084), CO (r = −0.038) and SO_2_ (r = −0.034) (*p* < 0.05). (3) The random forest regression model had the best fitting effect, while the BP neural network model exhibited the best prediction. (4) The validation set of the BP neural network model, including average daily temperature, sunshine hours and PM_10_, showed the lowest root mean square error, mean absolute error and mean absolute percentage error, followed by support vector regression. Conclusions: The prediction trend of the BP neural network model, including average daily temperature, sunshine hours and PM_10_, successfully mimics the actual incidence, and the peak incidence highly coincides with the actual aggregation time, with a high accuracy and a minimum error. Taken together, these data suggest that the BP neural network model can predict the incidence trend of tuberculosis in Changde City.

## 1. Introduction

Tuberculosis (TB) is a chronic infectious disease caused by Mycobacterium Tuberculosis that infects many organs, the most common of which is pulmonary Tuberculosis [1]. Tuberculosis patients often have a similar history of tuberculosis contact, and tuberculosis expeller is an unignorable source of infection [2]. Mycobacterium tuberculosis spreads mainly through the air because it can be suspended in the nuclei of droplets discharged by patients who cough or sneeze, and it can infect healthy people when inhaled. Tuberculosis is a disease with a long history. In 1882, Dr. Robert Koch first detected the bacterium associated with tuberculosis and named it Mycobacterium tuberculosis [3]. In 1993, the World Health Organization (WHO) proposed that tuberculosis had become a global public health problem and declared the “global tuberculosis emergency”. Since 1997, the WHO has been releasing statistics on the global tuberculosis report every year and makes them publicly available. According to the *Global Tuberculosis Report 2021* published by the WHO [4], an estimated 9.9 million people worldwide were infected with tuberculosis in 2020, which is equivalent to 127 cases per 100,000 people. In 2019, TB was the 13th leading cause of global death and the first leading cause of death from a single infectious disease.

There are many factors influencing the incidence of tuberculosis. Studies have shown that meteorological factors [5], social and economic factors [6], and the geographical ecological environment [7] have been closely associated with the incidence of tuberculosis. Changes in meteorological factors will affect the pattern and burden of tuberculosis. Currently, a few studies have shown that average temperature, air pressure, relative humidity, precipitation and sunshine duration all impact the incidence of tuberculosis [8,9,10]. For example, Bie, S believed that Mycobacterium tuberculosis was more likely to survive in an environment of high humidity and precipitation, but not in an environment of high temperature and pressure. UV rays can cause damage to skin and eyes, so prolonged sun exposure can lead to weakened immunity and tuberculosis infection [8]. However, Li, Z believed that temperature affects the probability of TB transmission by changing the indoor/outdoor activity time of TB-susceptible people and TB-infected people. A relatively high wind speed promotes the spread of mycobacterium tuberculosis in the air, thus increasing the risk of TB [9]. Air transmission is the main avenue of tuberculosis infection; patients are infected via tuberculosis pathogens in droplets spread into the air through coughing or sneezing. Exposure to high levels of air pollutants damages the mucous membranes of the upper respiratory tract, which are the first line of defense against mycobacterium tuberculosis infection. In addition, continued exposure to air pollutants decreased the expression levels of interferon-γ (IFN-γ) and tumor necrosis factor-α (TNF-α), which play an important role in the fight against mycobacterium tuberculosis infection [9]. One study found that exposure to ambient air pollution (PM_2.5_, PM_10_, O_3_ and CO) significantly increased the risk of drug-resistant tuberculosis [11], and a recent systematic review and meta-analysis study confirmed that long-term exposure to PM_10_, SO_2_ and NO_2_ significantly increased the incidence of tuberculosis [12]. These studies provide EBM (evidence-based medical) evidence for the influence of air pollution factors on the incidence of pulmonary tuberculosis.

The influence of meteorological factors and air pollutants on the incidence of infectious diseases and the establishment of prediction models based on them have become a research hotspot in the field of epidemiology. On the basis of traditional time series analysis, many studies have begun to consider the influence of meteorological factors in different regions, especially temperature and precipitation, on the incidence of tuberculosis, and used them as independent variables to establish the incidence prediction model of tuberculosis [13,14,15]. At present, there are two kinds of common infectious disease prediction models: the first is the traditional mathematical prediction model, such as autoregressive integrated moving average model (ARIMA), regression prediction model, exponential smoothing model, etc.; the second is prediction models based on machine learning, such as support vector machine (SVM), random forest, and the BP artificial neural network model. [16]. Although traditional mathematical prediction models have been relatively mature in the prediction of infectious diseases, different models are suitable for different data characteristics, and each type of infectious disease prediction model has its own advantages and disadvantages. The biggest shortcoming of the traditional mathematical prediction model is that it cannot extract a nonlinear relation in a time series [17]. Therefore, the machine learning prediction model, which has good performance in dealing with nonlinear relations in time series, has been gradually applied to the modeling of infectious diseases through its special algorithms and advantages [18,19].

To sum up, considering the nonlinear relation between influencing factors and the incidence of tuberculosis, this research intends to analyze the tuberculosis notification data from the period of 2010–2021 from Changde City, Hunan province as well as the meteorological and air pollution data. It also intends to describe the occurrence characteristics of the tuberculosis and epidemic trend and explore the correlation between tuberculosis disease and meteorological factors and air pollutants. A machine learning algorithm (support vector regression, random forest regression and BP neural network) was used to construct the daily incidence prediction model of tuberculosis, based on meteorological factors and air pollutants. The mean absolute error (MAE), root mean square error (RMSE) and mean absolute percentage error (MAPE) were used to evaluate the prediction effect of each model; this was conducted to explore the best prediction model for tuberculosis and provide a basis for the prediction and early warning model construction of infectious diseases.

## 2. Materials and Methods

### 2.1. Data Collection

#### 2.1.1. Study Area

Changde City (28°~31° N, 110°~113° E), located in the north of Hunan Province, China, is not only an important node city of the Yangtze River Economic Belt, but an indispensable part of the Dongting Lake Ecological Economic Zone. With a total area of 18,200 square kilometers, the city has jurisdiction over 9 districts, counties (cities) and 5 administrative districts. According to the results of the seventh national census, the permanent population of the city is 5,279,102 people [20]. Changde City is inclined from west to east, with diverse landforms, a highly developed water system, numerous rivers, and continental and monsoon climate characteristics that are obvious, with “warm climate, four distinct seasons; The heat is sufficient and the rain is concentrated; Spring temperature variable, summer and autumn drought; The cold period is short and the summer heat period is long” [21].

#### 2.1.2. Data Sources

In this study, the daily TB notifications in Changde City from 1 January 2010 to 31 December 2021 were obtained from the Disease Surveillance Information Report Management System of Changde City Center for Disease Control and Prevention. A total of 61,018 patients were collected with valid information, which were presented in the form of individual cases, including gender, age, occupation type, date of diagnosis, whether they were severe patients and diagnosis results, etc. The diagnosis of tuberculosis was based on the WS 288-2017 versions of the Diagnostic Criteria for Tuberculosis released by the Chinese Health and Family Planning Commission [1]. The case report registration information form was filled in by Changde City disease control professionals and inputted into the infectious disease information report management system within 24 h. Informed consent was not required because the incidence data were from the disease surveillance information report management system.

Meteorological data from 1 January 2010 to 31 December 2021 were downloaded from the National Climate Data Sharing Center (http://data.cma.cn/ (accessed on 20 May 2022)). Meteorological variables included daily average temperature (°C), maximum temperature (°C), minimum temperature (°C), relative humidity (%), air pressure (kpa), precipitation (mm), average wind speed (m/s) and sunshine hours (h).

Air pollutants were downloaded from the Department of Ecology and Environment of Hunan Province (http://sthjt.hunan.gov.cn/ (accessed on 20 May 2022)). The air quality factors involved included particulate matter with an aerodynamic diameter of less than 2.5 μm (PM_2.5_, μg/m^3^), particulate matter with an aerodynamic diameter of less than 10 μm (PM_10_, μg/m^3^), ozone (O_3_, μg/m^3^), sulfur dioxide (SO_2_, μg/m^3^), nitrogen dioxide (O_2_, μg/m^3^) and carbon monoxide (CO, mg/m^3^).

Demographic data were downloaded from Hunan Statistical Yearbook.

### 2.2. Statistical Analysis

#### 2.2.1. Descriptive Analysis

The daily TB notifications in Changde City from 2010 to 2021 were collected to describe the basic epidemiological characteristics of the tuberculosis cases.

#### 2.2.2. Correlation Analysis

Spearman rank correlation analysis is often used to analyze the correlation between various factors and the incidence of infectious diseases. The applicable conditions are as follows: (1) original data are represented by grades; (2) data do not obey bivariate normal distribution, so it is not suitable for product difference correlation analysis; (3) the overall distribution is unknown. After pre-analysis, this study found that tuberculosis incidence data did not conform to normal distribution, so non-parametric Spearman rank correlation analysis was used to detect the relationship between various meteorological factors, air pollutants and TB notifications.

#### 2.2.3. Support Vector Regression (SVR)

Support vector regression is an application of a support vector machine in regression. Based on elegant mathematical theories, it aims to find an optimal plane in a multi-dimensional space that can divide all sample units into two categories [22]. This plane should make the distance between the nearest points in the two categories as large as possible, and the segmented hyperplane is located in the middle of the distance [23]. For regression problems, support vector regression assumes that the maximum relative error between the predicted value and the actual value is the loss function ε. When the difference between the two values is less than ε, the loss does not need to be calculated. If the difference between the two values is greater than ε, the loss needs to be calculated to obtain the optimal solution [24]. Support vector regression modeling steps are as follows: (1) Data normalization and splitting—after normalization, the data from all the years are divided into training samples and validation samples, with the data from 2010 to 2020 as the training set and the data from 2021 as the validation set. (2) Select the optimal kernel function according to the data type and model comparison to establish the training model. (3) The optimal parameters of the training model are independently searched to establish the optimal model through cross validation and hyperparameter optimization. (4) Establish the optimal regression model and evaluate the accuracy of the model through the prediction results of the training set. (5) Prediction application: each sample of the validation set is entered into the model, and the prediction result of the validation set is finally obtained. The predicted value is compared with the actual value to verify the accuracy of the model.

#### 2.2.4. Random Forest Regression (RFR)

Random forest is a machine learning algorithm proposed by American scientist Leo Breiman [25] in 2001. It is an ensemble learning model based on a decision tree as a basic classifier, and contains multiple decision trees trained by Bagging ensemble learning technology [26]. For each sample unit, all decision trees predict and output it in turn, and the mean of the prediction results of these decision trees is the same value predicted by the random forest. That is, when the samples to be predicted are inputted, the final output results are jointly determined by the output results of all decision trees according to the principle of least mean square deviation. The steps of the random forest algorithm are as follows: (1) firstly, all samples X are divided into training set T and verification set V, assuming that there are N samples and M variables in the training set. (2) N sample units with the same size as the training set are randomly selected from the training set, and each sample unit builds a corresponding decision tree to generate a large number of decision trees (Bagging thought). (3) When the decision tree is segmented, M variables are randomly selected from each node as candidate variables, and then the most appropriate variable is selected as the split node (feature subspace idea). (4) All decision trees are completely generated without pruning, and a random forest is established. (5) Test samples: each new observation variable in the validation set is entered into each decision tree in the random forest for regression output, and the mean output of each decision tree is taken as the final result according to the principle of least mean square deviation. The schematic diagram of the random forest regression structure is shown in Figure 1.

#### 2.2.5. Back Propagation Neural Network (BPNN)

The Back propagation neural network, also known as the BP neural network, is one of the most widely used artificial neural networks. It was formally proposed by a group of scientists led by Rumelhart and McCelland in 1985 [27]. The BP network is a kind of supervised machine learning algorithm, which has a strong self-learning and adaptive ability. Its structure is mainly divided into multi-layer feedforward networks, namely input layer, hidden layer and output layer. The neurons in the input layer, hidden layer and output layer of a BPNN are not connected with each other, but each layer is connected with each other by using different functions. A Sigmoid function (such as logistic) is often used as a transfer function from the input layer to the hidden layer, and a linear transfer function is often used to connect the hidden layer to the output layer. The steps of establishing a BP neural network are as follows: (1) Data normalization and splitting—same as SVR. (2) Determine the network structure and setting function—the network structure is set as N-M-Q, where N is the number of nodes in the input layer, M is the number of neurons in the hidden layer, and Q is the number of neurons in the output layer. The above indexes are assigned according to the research purpose, and the connection function of the hidden layer and the output layer and the learning function of the network are also set. (3) Establish and test the neural network—train the network with training samples and test the network with test samples. (4) Determine the best network—perform 2–3 steps repeatedly, select the best network model by comparing the evaluation indexes of the test set, and make a prediction and an evaluation.

#### 2.2.6. Model Evaluation

The evaluation indexes of the prediction model in this study include root mean square error (RMSE), mean absolute percentage error (MAPE) and mean absolute error (MAE):$$\mathrm{RMSE}=\sqrt{\frac{1}{n}\sum_{i=1}^{n}{\left(Y_{i}-{\hat{Y}}_{i}\right)}^{2}}\;\mathrm{MAPE}=\frac{1}{n}\frac{\sum {\left|{\hat{Y}}_{i}-Y_{i}\right|}^{}}{Y_{i}}\;\mathrm{MAE}=\frac{1}{n}\sum_{i=1}^{n}\left|Y_{i}-{\hat{Y}}_{i}\right|$$

*Y_i_* is the actual value, $\bar{Y}$ is the mean of the actual value, and $\hat{Y}$ is the predicted value.

### 2.3. Software Applications

Descriptive analysis and correlation analysis were conducted with IBM SPSS 20.0 (International Business Machines Corporation, Armonk, NY, USA). The prediction model was built by R4.1.3 software (R Foundation for Statistical Computing, Vienna, Austria), in which the packages of “e1071”, “randomForest” and “neuralnet” were used to build the support vector regression model, the randomForest regression model, and the BP neural network model, respectively. Statistical test level was α = 0.05.

## 3. Results

### 3.1. Basic Characteristics of Tuberculosis Incidence Data, Meteorological Data and Air Pollutants Data

From 2010 to 2021, there were 61,018 newly diagnosed tuberculosis patients in Changde City, including 43,720 males and 17,298 females, with an annual incidence rate of 89.18/100,000. The age of onset was mainly 16–59 years old, followed by over 60 years old. Farmers accounted for the largest proportion of patients, followed by household workers and unemployed groups (Table 1). Overall, the incidence of tuberculosis in Changde City showed a downward trend (Figure 2).

A total of eight meteorological factors and six ambient air pollutants were selected in this study, and their basic descriptions are shown in Table 2.

### 3.2. Correlation Analysis

The daily TB notifications in Changde City from 2010 to 2021 were positively correlated with average temperature (r = 0.231), maximum temperature (r = 0.194), minimum temperature (r = 0.165), sunshine hours (r = 0.329), PM_2.5_ (r = 0.097), PM_10_ (r = 0.215) and O_3_ (r = 0.084) (*p* < 0.05), and were negatively correlated with mean air pressure (r = −0.119), precipitation (r = −0.063), relative humidity (r = −0.084), CO (r = −0.038) and SO_2_ (r = −0.034) (*p* < 0.05). There was no significant correlation between the daily TB notifications and the mean wind speed and NO_2_ (*p* > 0.05). According to the correlation coefficients, there are strong correlations between mean temperature, maximum temperature, minimum temperature, mean air pressure and PM_2.5_, PM_10_ (|r| > 0.7, *p* < 0.05) (Table 3).

### 3.3. Support Vector Regression (SVR)

In this study, TB notifications from 2010 to 2020 were taken as the training set and TB notifications in 2021 as the validation set. In this study, we selected variables to construct two different models based on the correlation between the number of daily reports of tuberculosis and the meteorological factors and air pollutants in the same period. Model 1 (SVR1) included daily average temperature, relative humidity, precipitation, sunshine hours, PM_10_, SO_2_, CO and O_3_, which were correlated with the number of daily incidences as independent variables (excluded because of the collinearity between maximum temperature, minimum temperature, average air pressure, PM_2.5_ and the included variables). Model 2 (SVR2) excluded variables with correlation coefficients less than 0.09, and only included average daily temperature, sunshine hours and PM_10_ with correlation coefficients greater than 0.1 (when the absolute value of the correlation coefficient was less than 0.09, it was considered to have no correlation [28]). Meanwhile, weekly variables (0 = weekend, 1 = working day) and holiday variables (0 = holiday, 1 = non-holiday) were substituted as covariates, and the two above models were used to predict the trend of the daily incidence of tuberculosis in Changde City in 2021.

In machine learning models, the support vector machine and the neural network, which are solved by the gradient descent method, usually need to normalize the data first, which aims to bring the data of different dimensions to within the same range so that they can more quickly find the optimal solution through gradient descent and can eliminate the influence of characteristics between the dimensional data. The normalization formula is as follows (*X*_norm_ is the normalized value): $X_{\mathrm{norm}}=\frac{X_{i}-X_{\min(i)}}{X_{\max(i)}-X_{\min(i)}}$. In the process of support vector regression modeling, the most important problem is the actual parameter setting. During modeling, the optimal parameters of the optimal model can be independently found through ten-fold cross-validation and the tune.svm function. The final basic parameter setting is as follows (Table 4):

The predicted results are shown in Figure 3. The incidence trend predicted by support vector regression is roughly consistent with the actual incidence trend. The accuracy of model SVR1 in January, March and April is higher than that of model SVR2, and the prediction accuracy of SVM2 in other months is higher than that of model SVR1. The daily predicted values were counted on a monthly basis, and the results were shown in Figure 4. It can be seen that the number and trend of the predicted incidence of SVR2 were similar to the reality, and the peak of the incidence was accurately predicted in June, with small data differences and high accuracy. On the whole, the accuracy of SVR2 was better than that of SVR1.

### 3.4. Random Forest Regression (RFR)

According to Leo Breiman’s suggestion, the number of variables (m) at each node in the random forest is generally 2~M, randomly selected from the total characteristic variable (M). In this study, model establishment and independent variable selection are the same as support vector regression. Therefore, the Mtry of RFR1 can be 2 to 10, and the Mtry of RFR2 can be 2 to 5. When selecting the optimal parameters, the combination of Mtry and Ntree finally selects the following parameters according to the principle of minimum mean square error (MSE) through continuous attempts (Table 5):

When the Mtry of RFR1 is 3 and Ntree is 1000, the minimum MSE of the model validation set is 27.67. When Mtry = 2 and Ntree = 1000 for RFR2, the minimum MSE of the model validation set is 24.01. The optimal parameters were selected to construct the final model and predict the daily incidence of pulmonary tuberculosis in Changde City in 2021. The results are shown in Figure 5. The fitting effect of random forest regression is good, and it can accurately predict the incidence trend of pulmonary tuberculosis in Changde City; the prediction accuracy of RFR2 at the low incidence value is better than that of RFR1. The daily value prediction number was counted by month, as shown in Figure 6. Both RFR1 and RFR2 could accurately predict the incidence trend of tuberculosis, but on the whole, RFR2 had better prediction performance, and the predicted value was more consistent with the actual incidence value and with higher accuracy.

### 3.5. Back Propagation Neural Network (BPNN)

The BP neural network model mainly contains a multi-layer feed-forward network with an input layer, hidden layer and output layer, and each characteristic variable included in the model is taken as a node of the input layer. In this study, two BP neural network regression models were established according to the relationship between the daily incidence of tuberculosis, meteorological factors and air pollutants in the same period (model establishment and independent variable selection were the same as support vector regression). We used an empirical formula to preliminarily determine the number of neurons in the hidden layer: $M=\sqrt{m+n}+a$, where M is the number of neurons in the hidden layer, m is the number of neurons in the output layer, n is the number of neurons in the input layer, and a is a constant in the range of 1 to 10 [29]. Therefore, the number of nodes in the input layer and output layer of BP1 is 10 and 1, respectively, and the number of nodes in the hidden layer is 3–13. The number of nodes in the input layer of BP2 is 5, the number of nodes in the output layer is 1, and the number of nodes in the hidden layer is 2–12. Before fitting the model, the data should be normalized as in support vector regression (see support vector regression for details). The parameters of the model should be determined according to the minimum MSE of the fitted model, and the final parameter selection is shown in Table 6.

When the network structure is 10-3-1, the number of neurons in the hidden layer is 3, the threshold is set to 0.01, and the minimum MSE of the BP1 test set is 25.40. When the network structure is 5-5-1, the number of neurons in the hidden layer is 5, the threshold is set to 0.05, and the minimum MSE of the BP2 test set is 22.85. The model structure of the BP neural network is shown in Figure 7. The above two best models were used to predict the daily incidence of pulmonary tuberculosis in Changde City in 2021, and the results are shown in Figure 8. The prediction accuracy of BP1 in March and April is better than that of BP2; the prediction accuracy of BP2 in July, August and September is better than that of BP1; and the prediction accuracy of BP1 in other months is not significantly different from that of BP2. The predicted number of daily values was counted monthly, as shown in Figure 9. On the whole, BP2 was much more accurate in predicting the incidence trend, and the predicted value was closer to the actual incidence value, with higher accuracy.

### 3.6. Model Comparison

RMSE, MAE and MAPE were used to compare the SVR, RFR and BP neural network models. Table 7 shows the fitting and prediction effects of the six models discussed in this study. The random forest regression model (RFR1 and RFR2) has the best fitting effect, while the BP neural network model (BP1 and BP2) has the best prediction effect. The validation set of the BP neural network model (BP2), including average daily temperature, sunshine hours and PM_10_, has the lowest root mean square error, mean absolute error and mean absolute percentage error, followed by SVR2. The support vector regression model (SVR1) that included average daily temperature, precipitation, relative humidity, sunshine hours, PM_10_, O_3_, CO and SO_2_ had the highest root mean square error, mean absolute error and mean absolute percentage error. Figure 10 shows the prediction effect of each model after monthly statistics. After comparison, the prediction trend of BP2 is near the actual incidence, the peak of incidence highly coincides with the actual aggregation time, with high accuracy and minimum error. It is suggested that BP2 can be used to predict the incidence trend of tuberculosis in Changde City.

## 4. Discussion

In this study, Spearman rank correlation analysis was used to analyze the correlation between the daily TB notifications and meteorological factors and air pollutants in Changde City, Hunan Province. Meanwhile, a machine learning algorithm was utilized to construct a tuberculosis incidence prediction model based on meteorological and air quality. MAE, RMSE, and MAPE were performed to evaluate the prediction model. This study aimed to explore an accurate prediction model of tuberculosis incidence, predict the epidemic trend of tuberculosis cases, and provide reference for epidemic prevention and control departments.

From 2010 to 2021, the overall incidence of tuberculosis in Changde City, Hunan Province showed a downward trend. The number of male cases was larger than that of female cases, the age of onset mainly ranged from 16 to 59 years old, and the occupational population with the largest number of cases was farmers. It is very consistent with the epidemic characteristics of tuberculosis in other provinces of China. For example, from 2005 to 2018, 70.45% of the total cases of tuberculosis in Henan Province were male, the median age was 48 years old, and the number of farmers and herdsmen reached 81.5% [30]. Men may be more socially active and smoke more than women, which increases the risk of TB [31,32]. The low immunity of middle-aged and elderly people [33], low income of residents in rural areas and imperfect medical services [34] may lead to an increased risk of tuberculosis, suggesting that we should pay more attention to men, middle-aged and elderly people and rural areas, and make targeted prevention and control measures to improve the detection and control rate of tuberculosis.

A total of eight meteorological factors and six air pollutants were included in the correlation analysis, among which the daily TB notifications were positively correlated with the daily average temperature, maximum temperature, minimum temperature, sunshine hours, PM_2.5_, PM_10_ and O_3_, and negatively correlated with the daily average pressure, precipitation, relative humidity, CO and SO_2_. There was no significant correlation between the daily TB notifications and the average wind speed and NO_2_. Higher temperatures may be associated with longer sunshine hours, and a systematic review suggests that higher temperatures lead to air currents that are generally high, providing a favorable environment for TB transmission and suggesting that pathogens replicate more easily at higher temperatures [35]. Long-term exposure to air pollution may lead to an increase in the number of reported cases of tuberculosis [36,37]. Studies have shown that PM 10 is associated with tuberculosis with a positive sputum culture, and the severity of the lung lesions increases with the increase in PM_10_ level [38]. PM 2.5 and PM 10 are solid particles that can change the key components of the anti-mycobacterial host immune response [39], inhibit the natural defense barrier of the respiratory tract, cause oxidative stress in lung cells and increase the proinflammatory response [40].

Using models to predict tuberculosis incidence could help identify trends and provide the basis for disease warning. This study used correlation analysis results using support vector regression, random forest regression and BP neural network model to construct six tuberculosis disease prediction models, and it compared the prediction results of the six models. It was found that the MAE, RMSE and MAPE of the BP neural network model, including average daily temperature, sunshine hours and PM_10_, were all lower than those of the support vector regression model and random forest regression model. We assume that the BP neural network model has better predictive performance than SVR and RFR for tuberculosis incidence. Different prediction models for infectious diseases have obvious advantages and disadvantages, so it is crucial to select an appropriate prediction model according to its applicable conditions and sample data characteristics [16]. SVR is a nonlinear model that can map data into a high-dimensional space so as to find a more appropriate regression curve in the high-dimensional space through a kernel function [41]. SVR is highly inclusive of data and can use an ε insensitive loss function to perform linear regression in the high-dimensional feature space to reduce the complexity of the model. However, when it is applied to large sample data, it will consume a lot of machine memory and operation time [42]. Random forest is an ensemble learning method that obtains more accurate results through a large number of decision trees. When the number of decision trees is large, it requires a lot of time and space [43]. Random forest can process a large number of data samples without normalization of the data processing, and it has a strong ability to use data sets [44]. Therefore, random forest has a high degree of fitting to the training set. The BP network has a strong self-learning ability and a highly nonlinear mapping ability. It can automatically summarize the nonlinear function relationship between data by learning or training without any prior formula [45]. However, the BP neural network has many modeling parameters, so the determination of its structure is a major difficulty in the modeling process [46]. The BP neural network model is intuitive and has good prediction ability. It was the best prediction model for tuberculosis in this study, which suggests that the BP neural network model can be used as a method to predict the incidence trend of infectious diseases by using external factors such as weather and air pollutants.

However, some questions in this study still need more in-depth research. First, the incidence of tuberculosis may be related to the social, economic, cultural and individual conditions of the population. This study used the daily notification data of tuberculosis, so it could not obtain the social, economic and cultural data of the same period, and could not consider the individual conditions of the population. Second, studies have shown that extended exposure to air pollution has a long-term effect on lung health [47], and a brief exposure to a severe pollution event has had a long-term effect on health for nearly two decades [48]. However, this study did not consider the cumulative impact of long-term exposure to pollutants on the incidence of tuberculosis, and only took the immediate effect of meteorological and air pollutants on health as a predictor, which may have underestimated the impact of pollutants on the incidence of tuberculosis.

## 5. Conclusions

In summary, using the incidence data of tuberculosis in Changde City from 2010 to 2020 as the training set and the incidence data in 2021 as the validation set for prediction, it was found that the prediction accuracy of the BP neural network model is better than that of support vector regression and the random forest regression model, which can be applied to the incidence prediction of tuberculosis in Changde City, Hunan Province. This study also provides the scientific basis and reference information for the formulation of a tuberculosis prevention and control policy for public health departments.

## Figures and Tables

**Figure 1 ijerph-20-03910-f001:**
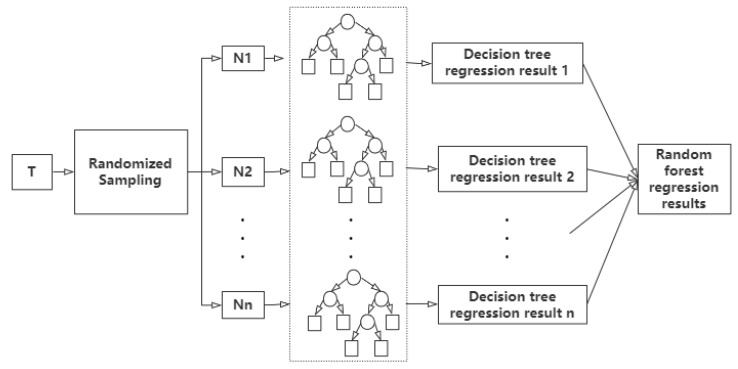
The schematic diagram of random forest regression structure.

**Figure 2 ijerph-20-03910-f002:**
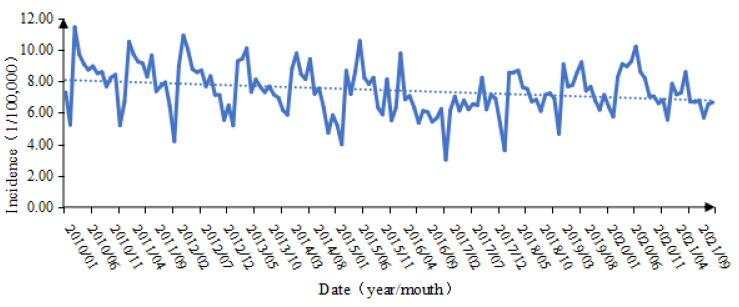
Time series of tuberculosis incidence in Changde City from 2010 to 2021.

**Figure 3 ijerph-20-03910-f003:**
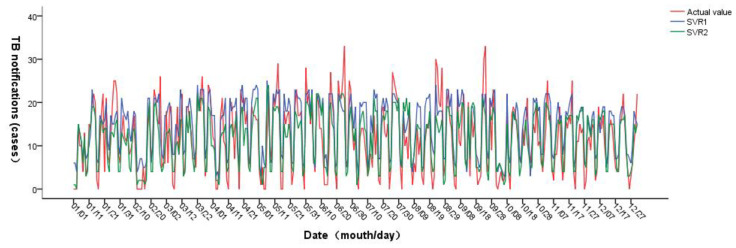
Prediction results of support vector regression.

**Figure 4 ijerph-20-03910-f004:**
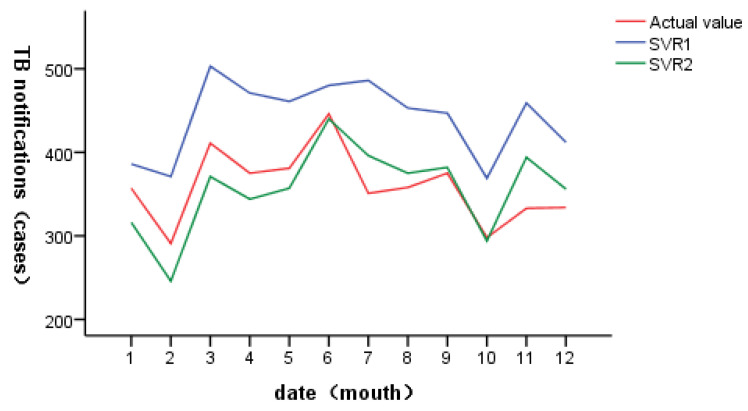
Prediction results of support vector regression (by monthly statistics).

**Figure 5 ijerph-20-03910-f005:**
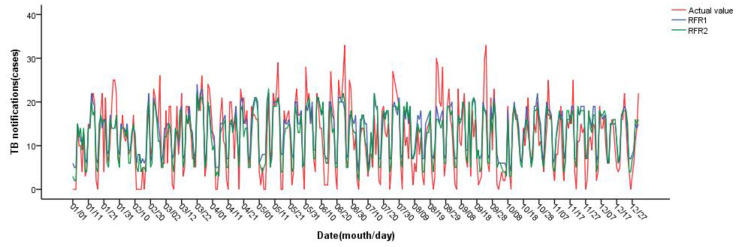
Prediction results of random forest regression.

**Figure 6 ijerph-20-03910-f006:**
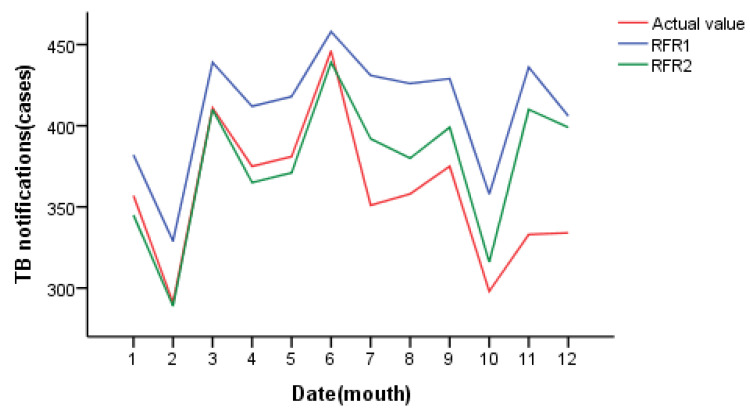
Prediction results of random forest regression (by monthly statistics).

**Figure 7 ijerph-20-03910-f007:**
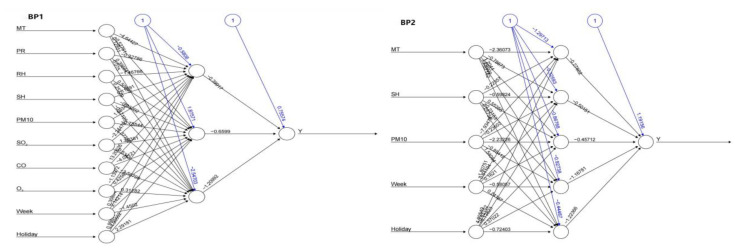
Structure diagram of BP neural network model (BP1/BP2).

**Figure 8 ijerph-20-03910-f008:**
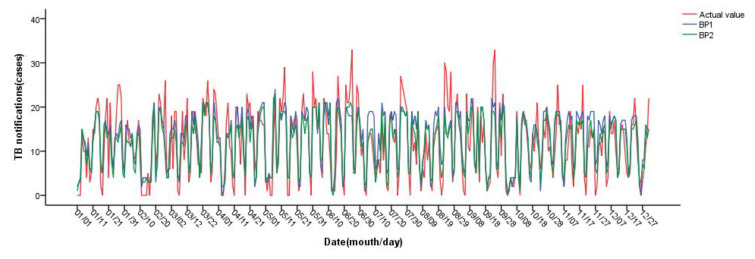
Prediction results of BP neural network.

**Figure 9 ijerph-20-03910-f009:**
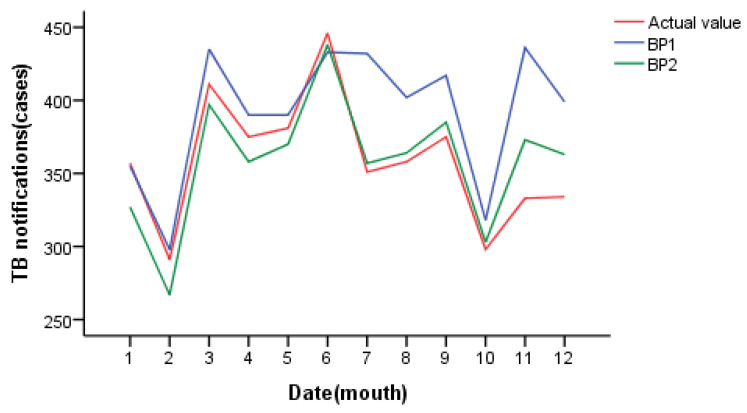
Prediction results of BP neural network (by monthly statistics).

**Figure 10 ijerph-20-03910-f010:**
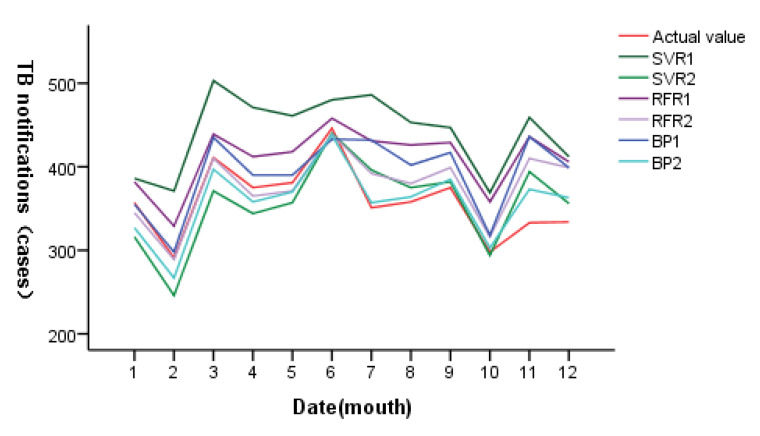
Prediction results of SVR, RFR and BP neural network prediction models (by monthly statistics).

**Table 1 ijerph-20-03910-t001:** The TB notifications in Changde City, Hunan Province, 2010–2021.

Variables	TB Notifications	Constituent Ratio (%)
Sex		
Male	43,720	71.7
Female	17,298	28.3
Age group		
≤15 years	346	0.6
16–59 years	32,735	53.6
≥60 years	27,937	45.8
Occupations		
Famers	49,053	80.4
Housework and unemployment	3186	5.2
Workers	2139	3.5
Retired persons	1839	3.0
Students	1749	2.9
Business services	885	1.5
The cadre staff	421	0.7
Teachers	273	0.4
The medical staff	201	0.3
Food and beverage industry	145	0.2
Public place attendant	69	0.1
Seafarers or long-distance drivers	78	0.1
Fisherman	41	0.1
Childcare workers and nannies	4	0.0
Unknown	250	0.4
Other	685	1.1

**Table 2 ijerph-20-03910-t002:** Basic descriptions of meteorological factors and ambient air pollutants in the Changde City from 2010 to 2021.

	Mean ± SD	Min	P5	Median	P95	Max
Meteorological factors						
MT (°C)	17.47 ± 8.78	−3.83	3.34	18.06	30.61	34.83
MAT (°C)	21.38 ± 8.28	−3.28	5.50	21.89	35.61	40.39
MIT (°C)	13.78 ± 8.35	−7.22	0.41	14.22	26.22	30.50
RH (%)	74.83 ± 12.12	23.71	52.59	75.99	92.42	99.07
PR (mm)	4.19 ± 10.32	0.00	0.00	0.00	23.32	121.16
MAP (kpa)	100.03 ± 1.04	97.87	98.50	100.03	101.78	103.49
MWS (m/s)	2.37 ± 1.03	0.10	0.98	2.37	4.37	9.62
SH (h)	3.67 ± 2.04	0.22	0.54	3.68	6.92	7.95
Air pollutants						
PM_2.5_ (μg/m^3^)	67.16 ± 35.26	3.00	24.39	60.85	134.42	369.87
PM_10_ (μg/m^3^)	92.81 ± 47.91	8.00	34.24	84.41	183.37	499.84
SO_2_ (μg/m^3^)	15.56 ± 5.37	4.00	7.47	15.17	25.33	43.84
NO_2_ (μg/m^3^)	19.23 ± 8.10	3.57	8.97	18.10	33.06	89.42
CO (mg/m^3^)	0.71 ± 0.32	0.16	0.33	0.65	1.23	8.77
O_3_ (μg/m^3^)	61.35 ± 22.46	2.80	28.75	58.46	101.99	151.12

MT: mean temperature; MAT: mean maximum temperature; MIT: mean minimum temperature; RH: relative humidity; PR: precipitation; MAP: mean air pressure; MWS: mean wind speed; SH: sunshine hours; PM_2.5_, PM_10_, SO_2_, NO_2_, CO are the 24 h average concentration, and O_3_ is the 8 h average concentration.

**Table 3 ijerph-20-03910-t003:** Spearman’s correlation coefficient matrix of daily TB notifications with meteorological factors and air pollutants in the same period.

Variables	TB Notifications	MT (°C)	MAP (kpa)	MWS (m/s)	MAT (°C)	MIT (°C)	PR (mm)	RH (%)	SH (h)	PM_2.5_ (μg/m^3^)	PM_10_ (μg/m^3^)	O_3_ (μg/m^3^)	CO (mg/m^3^)	SO_2_ (μg/m^3^)	NO_2_ (μg/m^3^)
TB notifications	1														
MT (°C)	0.231 *	1													
MAP (kpa)	−0.119 *	−0.754 *	1												
MWS (m/s)	−0.02	0.051 *	−0.308 *	1											
MAT (°C)	0.194 *	0.953 *	−0.737 *	0.052 *	1										
MIT (°C)	0.165 *	0.96 *	−0.772 *	0.043 *	0.928	1									
PR (mm)	−0.063 *	−0.112 *	−0.076 *	0.141 *	−0.192 *	−0.03	1								
RH (%)	−0.084 *	−0.047 *	−0.202 *	0.01	−0.152 *	0.062 *	0.629 *	1							
SH (h)	0.329 *	0.627 *	−0.385 *	−0.01	0.637 *	0.496 *	−0.378 *	−0.423 *	1						
PM_2.5_ (μg/m^3^)	0.097 *	−0.5 *	0.492 *	−0.068 *	−0.482 *	−0.52 *	−0.093 *	−0.242 *	−0.183 *	1					
PM_10_ (μg/m^3^)	0.215 *	−0.475 *	0.484 *	−0.069 *	−0.48 *	−0.519 *	−0.081 *	−0.22 *	−0.115 *	0.879 *	1				
O_3_ (μg/m^3^)	0.084 *	0.516 *	−0.396 *	0.121 *	0.526 *	0.508 *	−0.066 *	−0.116 *	0.376 *	−0.161 *	−0.17 *	1			
CO (mg/m^3^)	−0.038 *	−0.591 *	0.441 *	−0.105 *	−0.525 *	−0.614 *	−0.14 *	−0.048 *	−0.277 *	0.586 *	0.538 *	−0.34 *	1		
SO_2_ (μg/m^3^)	−0.034 *	−0.48 *	0.506 *	−0.104 *	−0.439 *	−0.519 *	−0.246 *	−0.311 *	−0.155 *	0.537 *	0.518 *	−0.403 *	0.501 *	1	
NO_2_ (μg/m^3^)	−0.014	−0.372 *	0.269 *	−0.162 *	−0.303 *	−0.427 *	−0.236 *	−0.168 *	−0.069 *	0.323 *	0.305 *	−0.535 *	0.599 *	0.514 *	1

*: *p* < 0.05; MT: mean temperature; MAT: mean maximum temperature; MIT: mean minimum temperature; RH: relative humidity; PR: precipitation; MAP: mean air pressure; MWS: mean wind speed; SH: sunshine duration; PM_2.5_, PM_10_, SO_2_, NO_2_, CO are the 24 h average concentration, and O_3_ is the 8 h average concentration.

**Table 4 ijerph-20-03910-t004:** Support vector regression parameter selection.

Parameters	Ranges	SVR1	SVR2
Training set	2010–2021	2010–2020	2010–2020
Validation set	2010–2021	2021	2021
Kernel function	LK, PK, RBF, Sigmoid	RBF	RBF
Type	eps-regression	-	-
Cross validation	Ten folds cross validation	-	-
Cost	10^−10^~10^10^	1	1
Gamma	10^−6^~10^1^	0.1	0.2
Epsilon	0.01, 0.05, 0.1, 1	0.1	0.1

Kernel functions commonly used for support vector regression: LK—Linear Kernel; PK—Polynomial Kernel; RBF—Radial Basis Function; Sigmoid Kernel.

**Table 5 ijerph-20-03910-t005:** Random forest regression parameter selection.

Parameters	Ranges	RFR1	RFR2
Training set	2010–2021	2010–2020	2010–2020
Validation set	2010–2021	2021	2021
Mtry	2~10, 2~5	3	2
Ntree	200~2000	1000	1000

Mtry—Random forest node parameters; Ntree—Random forest decision tree number parameters.

**Table 6 ijerph-20-03910-t006:** Back Propagation Neural Network parameter selection.

Parameters	Ranges	RFR1	RFR2
Training set	2010–2021	2010–2020	2010–2020
Validation set	2010–2021	2021	2021
Hidden	3~13, 2~12	3	5
Threshold	0.01, 0.05, 0.1	0.01	0.05

**Table 7 ijerph-20-03910-t007:** Comparison of SVR, RFR and BP neural network prediction models.

Models	Training Set			Validation Set		
	RMSE	MAE	MAPE	RMSE	MAE	MAPE
SVR1	5.81	4.29	0.52	5.72	4.56	0.67
SVR2	6.12	4.62	0.55	4.84	3.70	0.50
RFR1	0.73	2.08	0.3	5.26	4.22	0.67
RFR2	4.73	3.64	0.46	4.90	3.84	0.57
BP1	6.09	4.69	0.60	5.04	3.92	0.56
BP2	6.16	4.74	0.60	4.78	3.67	0.48

RMSE—root mean square error; MAE—mean absolute error; MAPE—mean absolute percentage error.

## Data Availability

The datasets used and/or analyzed in this study are available from the corresponding author on reasonable request.

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
