# Peer review of "Machine Learning Prediction Model of Tuberculosis Incidence Based on Meteorological Factors and Air Pollutants"

_ijerph, 2023, doi:10.3390/ijerph20053910_

Round 1
Reviewer 1 Report
The paper makes a marginal contribution. The techniques employed are really trivial. Significant improvements are required in the writing and presentation, particularly the figures and tables.
Author Response
Thank you for your review and comments. This study focuses on the application of machine learning algorithms to explore the impact of meteorological factors and air pollutants on the incidence of tuberculosis. We will make improvements in writing and presentation, especially in figures and tables. If you have any better suggestions, we hope we can get your specific advice at your convenience.
Reviewer 2 Report
The entire text needs to be revised regarding some inaccuracies in punctuation. Sometimes there are extra spaces or capitalization after a colon, for example. Please check.
In the introduction, the aim is missing. You describe what and how you will do but not why.
In section 2.2.4 in line 184 the training set was called “N” but the samples were also called “N”. It would be better to denote them by different letters. In addition, it might be useful to associate an explanatory image to better understand the organization of the groups.
Figure 1 is too large for the layout of the article. It appears cropped.
In Table 1, summing the groups in the variables “sex” and “age” gives a total of 61.018. In the variable "occupations" it is 60.975, 43 less. Please verify the numbers.
Please check whether from line 311 to 322 the uppercase and lowercase letters associated with "try" and "tree" are put in the correct mode.
In table 6 you used for the "hidden" parameter the slash to separate the values while for the "threshold" parameter the comma. I recommend using the same symbol.
In Figure 6, it is superfluous to denote the two networks with the letters A and B since they are already named BP1 and BP2.
Author Response
Dear reviewer:
thank you very much for your review. We have replied to your questions and comments one by one, and modified them in the paper. The details are as follows:
#1: The entire text needs to be revised regarding some inaccuracies in punctuation. Sometimes there are extra spaces or capitalization after a colon, for example. Please check.
Response: Thanks for your review and suggestions. We have corrected some punctuation inaccuracies in the article and checked the whole text.
#2: In the introduction, the aim is missing. You describe what and how you will do but not why.
Response: Thank you for your advice. In this study, Changde City, Hunan Province, China was selected as the research area. Machine learning algorithms (support vector regression, random forest regression and BP neural network) were used to construct a daily incidence prediction model of tuberculosis based on meteorological factors and air pollutants. Mean absolute error (MAE), root mean square error (RMSE) and mean absolute percentage error (MAPE) were used to evaluate the prediction effect of each model. The purpose of this study was to explore the best prediction model for tuberculosis and provide a basis for the construction of prediction and early warning model for infectious diseases. The above content has been added in the article. (Line 100-106)
#3: In section 2.2.4 in line 184 the training set was called “N” but the samples were also called “N”. It would be better to denote them by different letters. In addition, it might be useful to associate an explanatory image to better understand the organization of the groups.
Response: Thank you very much for your advice. We have called the training set "T" and the sample "N". As you suggested, we have added the structure diagram of random forest regression model in the paper, hoping that it can make you understand. (Figure 1).
#4: Figure 1 is too large for the layout of the article. It appears cropped.
Response: Thanks for your suggestion, we have changed Figure 1 to PNG image format to make the picture display more complete.
#5: In Table 1, summing the groups in the variables “sex” and “age” gives a total of 61.018. In the variable "occupations" it is 60.975, 43 less. Please verify the numbers.
Response: Thank you very much for your reminding. We are very sorry that we recorded the wrong data of a group of farmers due to our negligence. We have checked and corrected the data.
#6: Please check whether from line 311 to 322 the uppercase and lowercase letters associated with "try" and "tree" are put in the correct mode.
Response: Thank you for reminding us. We have checked the letters related to "try" and "tree" in lines 311-322 and unified them into "Mtry" and "Ntree" in the text.
#7: In table 6 you used for the "hidden" parameter the slash to separate the values while for the "threshold" parameter the comma. I recommend using the same symbol.
Response: Thanks for your careful review, we have modified Table 6 uniformly to use comma separated values.
#8: In Figure 6, it is superfluous to denote the two networks with the letters A and B since they are already named BP1 and BP2.
Response: Thanks to your suggestion, we have removed A and B from Figure 6.
Thank you very much for your valuable comments and suggestions. These comments are very valuable and helpful in revising and improving our paper. We hope our modification can meet the requirements of reviewers and IJERPH, and we wish you success in your work and good health.
Reviewer 3 Report
It is an interesting study on an important topic. The presentation is lacking clarity and context however, particularly in the introduction. The authors should expand their discussion and frame their study in the existent literature a bit better. I have a number of comments and suggestions that I think would considerably strengthen the paper:
1. It is unclear what the causal pathways are between the meteorological and environmental factors and the incidence of TB. The authors should briefly explain the results of the existent literature in terms of what the hypothesized causal chains are. It is not clear to the reader why temperature, pollutants, etc. correlate well with the incidence of TB. There is a bit of text on these issues in the discussion section, but some of these need to be presented in the introduction when citing the related literature so that the reader forms a general idea about what is going on from the very beginning.
2. I am particularly concerned about the linkages with the pollution variables. The author mention a couple of studies that show correlations between pollution exposure and TB, but once again it is not clear what is the causal pathway. I suspect that the causal pathway involves lung health, in the sense that pollution reduces lung health and this makes people more likely to get infected and develop TB. However if that is the case, contemporaneous pollution may not be the best measure to use. Lagged pollution (or a yearly average) may be more appropriate since it is this long-term exposure to pollution that hurts lung health and makes more likely that people contract and develop TB.
3. Somewhat related to this, there are a number of recent studies that show causal evidence of a persistent link between pollution exposure and lung health. For instance Kim et al. (2017) in Economics and Human Biology and Kim and Radoias (2022) in International Journal of Environmental Research and Public Health show that pollution exposure has very persistent effects on lung health. The authors should at a minimum discuss these implications. Even if pollution is low currently, high pollution levels in the past may result in low population lung health and high incidence of TB. To investigate this possibility, the authors could for instance show us the dynamics of the pollution variables over time and control for past levels of pollution in the empirical models.
4. Overall, I would like a better distinction between the two distinct channels through which TB incidence can be affected: on one hand there are factors that make the TB bacterium easier to spread and on the other hand there are factors that make the local population more susceptible to getting infected. Note that not everyone who inhales the bacterium gets infected. Certain people are more at risk that others. For instance Narasimhan et al. (2013) in Pulmonary Medicine have a nice summary of these risk factors.
5. I honestly find it a bit unconvincing to predict current TB incidence using current meteorological and environmental factors since it takes some time from the spread of the bacterium to the infection of the individual and the case reporting. I get that many of these variables are serially correlated and so to some degree, current values are strongly in line with past values, but the models may gain power if past values are incorporated, especially when there are significant seasonal changes.
Author Response
Dear reviewer:
Thank you very much for your review. We have replied to your questions and comments one by one, and modified them in the paper. The details are as follows:
#1. It is unclear what the causal pathways are between the meteorological and environmental factors and the incidence of TB. The authors should briefly explain the results of the existent literature in terms of what the hypothesized causal chains are. It is not clear to the reader why temperature, pollutants, etc. correlate well with the incidence of TB. There is a bit of text on these issues in the discussion section, but some of these need to be presented in the introduction when citing the related literature so that the reader forms a general idea about what is going on from the very beginning.
Response: Thank you very much for your advice. We have included some possible causal chains in the introduction to summarize previous studies on why temperature, pollutants, etc. are closely related to the incidence of TB. (Line 66-81).
#2. I am particularly concerned about the linkages with the pollution variables. The author mention a couple of studies that show correlations between pollution exposure and TB, but once again it is not clear what is the causal pathway. I suspect that the causal pathway involves lung health, in the sense that pollution reduces lung health and this makes people more likely to get infected and develop TB. However if that is the case, contemporaneous pollution may not be the best measure to use. Lagged pollution (or a yearly average) may be more appropriate since it is this long-term exposure to pollution that hurts lung health and makes more likely that people contract and develop TB.
Response: Your point is very reasonable. Current studies have shown that exposure to high levels of air pollutants damages the mucous membranes of the upper respiratory tract, thus disrupting the first line of defense against mycobacterium tuberculosis infection. In addition, continued exposure to air pollutants reduced the expression levels of interferon-γ (IFN-γ) and tumor necrosis factor-α (TNF-α), which play an important role in the fight against mycobacterium tuberculosis infections. This lung damage may be related to long-term exposure to air pollutants. Our team also discussed the delayed influence of meteorological factors and air pollutants on the incidence of tuberculosis in the follow-up study. Due to the limited space of the study, this part will be presented separately. In this study, it is expected to find out whether short-term exposure to pollutants will have an impact on tuberculosis in the population, so as to screen out factors related to tuberculosis for prediction.
#3. Somewhat related to this, there are a number of recent studies that show causal evidence of a persistent link between pollution exposure and lung health. For instance Kim et al. (2017) in Economics and Human Biology and Kim and Radoias (2022) in International Journal of Environmental Research and Public Health show that pollution exposure has very persistent effects on lung health. The authors should at a minimum discuss these implications. Even if pollution is low currently, high pollution levels in the past may result in low population lung health and high incidence of TB. To investigate this possibility, the authors could for instance show us the dynamics of the pollution variables over time and control for past levels of pollution in the empirical models.
Response: Thank you very much for your advice. Our team is also exploring the delayed and cumulative effects of meteorological factors and air pollutants on the incidence of tuberculosis, and some achievements have been made so far. Continuous exposure to high levels of air pollutants such as PM2.5 or PM10 increases a population's cumulative risk of developing tuberculosis. Therefore, according to your suggestions, we also added the discussion of this part in this study. (Line496-503 )
#4. Overall, I would like a better distinction between the two distinct channels through which TB incidence can be affected: on one hand there are factors that make the TB bacterium easier to spread and on the other hand there are factors that make the local population more susceptible to getting infected. Note that not everyone who inhales the bacterium gets infected. Certain people are more at risk that others. For instance Narasimhan et al. (2013) in Pulmonary Medicine have a nice summary of these risk factors.
Response: Yes, we agree with you. Factors such as high temperatures and humidity may make it easier for mycobacterium tuberculosis to spread, while lung damage from air pollutants may make local populations more susceptible to infection. But we cannot deny that some people may have different individual effects that prevent them from becoming ill even if they are infected with Mycobacterium tuberculosis. However, this study did not fully focus on discovering the pathogenesis of tuberculosis, but hoped to predict the possibility of tuberculosis in the future by exploring the influence of meteorological factors and air pollutants on the incidence of tuberculosis, so as to give a better warning for the prevention and control of tuberculosis.
#5. I honestly find it a bit unconvincing to predict current TB incidence using current meteorological and environmental factors since it takes some time from the spread of the bacterium to the infection of the individual and the case reporting. I get that many of these variables are serially correlated and so to some degree, current values are strongly in line with past values, but the models may gain power if past values are incorporated, especially when there are significant seasonal changes.
Response: Thank you for your questions about this study. We have also discussed the questions you raised. Although current meteorological and environmental factors may have delayed and cumulative effects on the incidence of tuberculosis, we also found that current meteorological and air pollutants had nonlinear effects on the incidence of tuberculosis through Spearman rank correlation analysis. The machine learning algorithm can capture this nonlinear relationship well and predict the incidence of tuberculosis by using weather and air pollutants through self-learning method. Therefore, the three machine learning algorithms used in this study make use of the matching model of meteorological and environmental factors in the same period for prediction. At the same time, if past predictors are combined to explore the cumulative effect, how to determine when and how far to combine them? This issue will be further analyzed by our team in exploring the delayed and cumulative effects of meteorological factors and air pollutants on tuberculosis.
Thank you again for your valuable comments and suggestions. These comments are very valuable and helpful in revising and improving our paper. We hope our modification can meet the requirements of reviewers and IJERPH, and we wish you success in your work and good health.
Round 2
Reviewer 1 Report
I still have the same comment. The paper provides a very minor contribution. The techniques used are actually very simple and trivial. The findings are limited and not interesting nor useful due to the small size of the study region(Changde City, a small Chinese city from thousands of cities in China).
Reviewer 3 Report
The authors did a fairly good job in addressing my concerns. I still think the study can be improved, but I understand if the authors prefer to take on that task in a subsequent study that carefully considers the effects of long-term exposure.